# Prophylactic Effect of Fenestration on the Recurrence of Thoracolumbar Intervertebral Disc Disease in Dogs

**DOI:** 10.3390/ani12192601

**Published:** 2022-09-28

**Authors:** Afroditi E. Pontikaki, Kiriaki Pavlidou, Zoe Polizopoulou, Ioannis Savvas, George Kazakos

**Affiliations:** Companion Animal Clinic, School of Veterinary Medicine, Aristotle University of Thessaloniki, 546 27 Thessaloniki, Greece

**Keywords:** prophylactic fenestration, thoracolumbar disc disease, canine, dog, recurrence, intervertebral disc disease

## Abstract

**Simple Summary:**

The prophylactic effect of fenestration on the recurrence of thoracolumbar (TL) intervertebral disc herniation (IVDH) in dogs that have been surgically decompressed has been a topic of ongoing debate in veterinary medicine. The aim of this study was to systematically review the existing literature and critically evaluate the evidence behind the application of prophylactic fenestration on the recurrence of TL IVDH in dogs. PubMed, Web of Science and Scopus electronic databases were searched to collect relevant articles. Twenty-nine articles met the inclusion criteria and were assessed for scientific quality, treatment plan, and recurrence incidence. Five articles were selected for a meta-analysis to test if the recurrence differs in animals treated with or without prophylactic fenestration. In the light of the low scientific quality and the amount of published literature on the topic, further research is needed to robustly support the prophylactic effect of fenestration on the recurrence of TL IVDH in dogs.

**Abstract:**

This systematic review aimed to assess the effect of prophylactic fenestration (PF) on the recurrence of thoracolumbar (TL) intervertebral disc (IVD) disease in dogs. Three online databases were searched (Web of Science, MEDLINE via PubMed, SCOPUS), 115 relevant studies were thoroughly examined by the authors, 29 of which met the pre-defined inclusion criteria for this systematic review. Data about the initial treatment, the performance of PF, the incidence of recurrence, and the site of recurrence were extracted. Most of the studies were deemed to have serious to moderate risk of bias. Out of 5457 dogs, 1264 underwent prophylactic fenestration. A total of 504 cases of suspected or confirmed recurrence were recorded, in which 164 (32.54% of total recurrences and 11.02% of PF cases) were in dogs treated with PF. In order to perform quantitative analysis for the recurrence odds, we conducted a meta-analysis. Five studies were included that met the inclusion criteria. Despite a large number of relevant publications, the quality of the evidence they provide is low. This prevented us from reaching a definitive conclusion on the prophylactic effect of fenestration on recurrence in dogs surgically treated for TL IVDH.

## 1. Introduction

Thoracolumbar intervertebral disc herniation (IVDH) is reported as the most common IVDH in canine patients, with a ratio of 66–87% [1]. The intervertebral disc spaces of T12-T13 to L2-L3 have been shown to be at higher risk, especially the T12-T13 and T13-L1 [2,3,4,5,6,7]. Hansen type I disc extrusion is commonly seen with chondrodystrophic breeds and the Dachshund shows a particular high risk [1]. For a definitive diagnosis of TL intervertebral disc herniation (IVDH) and localization of the herniated disc, the gold standard of imaging studies is Magnetic Resonance Imaging (MRI) [1,8,9,10]. Surgical decompression, with hemilaminectomy being the preferred method, is invariably the treatment of choice when neurological deficits ensue [11]. Concomitant prophylactic fenestration of the adjacent not affected discs is not always performed as its efficacy as a prophylactic measure is questioned. In a survey conducted among board-certified surgeons and neurologists in order to gain information about the management of TL IVDH in everyday clinical practice, fenestration was performed by 69% of neurologists but only by 36% of surgeons. On the same questionnaire, 47% of the surgeons responded that they never or rarely performed fenestration in contrast to 12% of neurologists [12].

Fenestration involves the removal of the nucleus pulposus (NP) through a window in the annulus fibrosus (AF) of the intervertebral disc (IVD) [13]. It is a surgical method used for management of patients with TL IVDH, attempting not only treatment of acute pain but prophylaxis against early and late recurrence at the adjacent and other discs. Fenestration could be accomplished by manual (blade), power-assisted, percutaneous laser disc ablation (PLDA), cavitron ultrasonic surgical aspiration (CUSA), vacuum aspiration, chemonucleolysis and, most recently, discolysis by injection of radiopaque gelified ethanol [14,15,16,17]. The effectiveness of the procedure is based on the amount of the NP removed, although complete removal is not achievable, and it is correlated with a surgeon’s skills [18,19]. Fenestration seems to reduce the chance of further herniation of the prophylactically treated discs, especially in cases of mineralized discs. This favors Dachshunds especially, who are 10 times more prone to recurrence than other breeds, especially in adjacent mineralized discs [13,20,21,22,23,24]. However, there is no confirmed evidence that after fenestration the created window will remain open, or it will act as an alternate path [13,25]. Despite the fact that fenestrated discs appear to not herniate, relapse of disease seems to occur in the adjacent and/or further discs spaces [1]. Moreover, the increase in anesthesia and surgical time, the post-surgical exacerbation of the neurological deficits, the risk of complications, such as hemorrhages and iatrogenic pneumothorax, post-surgical morbidity (scoliosis, abdominal wall weakness) and discospondylitis are some of the main arguments against fenestration. Furthermore, PF has been identified as a predisposing factor of vertebral subluxation and instability, disc degeneration and persistent lameness or paresis on pelvic limbs (due to root trauma when fenestration is performed on L3 and on more caudal intervertebral disc spaces) [26]. Considering that it may only favor a few animals and comes with the risk of recurrence, the question if fenestration should be applied prophylactically as a routine remains unanswered.

This systematic review critically evaluates the existing literature and research data to determine if there is enough evidence to safely support the argument that fenestration decreases the risk of recurrence in thoracolumbar disc (TLD) disease in canine patients, or if further research and investigation are required.

## 2. Materials and Methods

The protocol of this systematic review was performed according to the guidelines of the Systematic Review Protocol for Animal Intervention Studies (SYRCLE).

Two research teams with two researchers each (group A: GK/AP; Group B: IS/KP) were set up to evaluate and assess the articles and conduct the systematic review (SR). Any authors of this review were excluded from the evaluation of any eligible study in which they were also authors.

### 2.1. Literature Search and Identification

The PICO (P: population, I: intervention, C: control, O: outcome) question of this research is if PF in TLD spaces, in dogs previously suffering from and treated for TLD herniation, decreases the risk of recurrence of TL IVD prolapse. Three electronic databases were used as literature sources to determine the incidence of recurrence through references in articles of interest: MEDLINE via PubMed, Web of Science and Scopus. The string sequence used in each database engine was: (thoracolumbar disc* OR thoracolumbar disk*) OR (fenestration OR ablation OR annulectomy OR discectomy) AND (canine* OR dog*) AND (recurrence).

### 2.2. Study Selection

Types of studies included in this review were cohort studies (retrospective, prospective and case series studies) and clinical studies (randomized clinical trials and blinded clinical trials) in English up to June 2022. Cadaveric studies, review studies and case reports were excluded from our research. The population investigated was dogs with TLD disease of all breeds and ages. Surgically treated dogs without concomitant PF when the relapse outcome was recorded were set as the control group, while the surgically treated animals with concomitant PF were defined as the intervention group. The surgical treatment identified was any surgical technique chosen for treatment of the affected disc. In this SR, the primary outcome established the number of dogs that experienced a recurrence episode. Related data were collected from the available follow-up (clinical examination from a veterinarian or telephone communication with the owners). Confirmed (by surgery or imaging study) and suspected (presentation suffering related clinical signs) recurrences were both included. Complications related with PF were the secondary outcome of interest. Unrelated to this review studies, duplicated research, studies unavailable in full texts and abstract only papers were excluded from this SR. Similarly, studies that did not fit the inclusion criteria were excluded. The PICO framework of the study featuring the inclusion criteria of the systematic review is demonstrated in Table 1.

### 2.3. Study Screening and Data Extraction

After screening the included studies at the title and abstract level from the two author groups, the data extracted from each research were: author, title, year, sample number, surgical treatment of choice, performance of PF, incidence of recurrence, site of recurrence (if mentioned in detail).

### 2.4. Quality Assessment

For the introduction, storage, analysis, and synthesis of data, a specialized software (Review Manager/RevMan Version 5.3. Copenhagen: The Nordic Cochrane Centre, The Cochrane Collaboration, 2014) was used. To evaluate the risk of bias on each of the included articles, two risk of bias assessment tools were used: ROBINS I for the cohort studies (retrospective studies and prospective studies/case series) with the recommended scale: Low–Moderate–Serious–Critical–No information. RoB2 was used for the clinical trials (randomized clinical trials, blinded clinical trials) with the recommended assessment scale: High–Low–With Some concerns in the RoB2 tool. Each tool was used according to the existing guidelines (https://www.riskofbias.info/, accessed on 23 September 2022). Quality reviews were independently conducted by the two research teams (GK/AP; IS/KP).

### 2.5. Data Analysis

Descriptive statistics is used to evaluate the results of our SR. Recurrence incidence among the total population examined, PF cases and Dachshunds were recorded. Additionally, the recurrent incidences, other than the primary surgical treated area (cranially and caudally), were recorded, as were areas of interest according to the literature. To evaluate the difference of the incidence of recurrence after PF, a meta-analysis was conducted. Included studies investigate the number of prophylactic fenestrations performed, recurrence rate and site of recurrence. Only the confirmed recurrence cases (by surgery or imaging study) were included to identify the incidence. A second meta-analysis was conducted to evaluate the difference of the incidence of recurrence after PF on adjacent disc(s). The selected inclusion criteria were the same as in the previous meta-analysis, with the addition of cases that experienced confirmed recurrence on disc(s) adjacent to a previously surgically treated area. RevMan software was used for the meta-analysis. The outcome was dichotomous and analyzed by the Mantel–Haenszel method with a random effects model. The I-squared index was employed to evaluate heterogeneity, with values: 0% to 40%: might not be important; 30% to 60%: may represent moderate heterogeneity; 50% to 90%: may represent substantial heterogeneity; 75% to 100%: considerable heterogeneity. Heterogeneity was calculated. Statistical significance was set to α = 0.05.

## 3. Results

As evidenced in the flow diagram (Figure 1), 818 articles were extracted from the selected electronic databases. After the removal of the duplicates, 648 articles remained. After screening the articles at title and abstract level from the two research groups (GK/AP; IS/KP), 115 were eligible for further screening at full text level. Twenty-nine articles met the inclusion criteria for the SR. From the 29 included studies, 27 were cohort studies and 2 were randomized clinical trials. Five of them were suitable for meta-analysis.

### 3.1. Characteristics of the Excluded Studies

Eighty-six studies were excluded from the SR for the following reasons:Articles not available in English (10) [3,27,28,29,30,31,32,33,34,35].Type of study: case report (7) [36,37,38,39,40,41,42]; cadaveric study (9) [16,17,25,43,44,45,46,47,48]; review study (9) [1,49,50,51,52,53,54,55,56]; questionnaire (1) [57].Recurrence was not investigated (35) [7,14,58,59,60,61,62,63,64,65,66,67,68,69,70,71,72,73,74,75,76,77,78,79,80,81,82,83,84,85,86,87,88,89,90].Recurrence was investigated only for the surgically treated site (3) [18,91,92].Abstract or full paper were not available (10) [2,93,94,95,96,97,98,99,100,101].Primary site of IVDH is not mentioned to evaluate the recurrence (1) [102].The intervention group of each recurrent dog is not specified (1) [103].

### 3.2. Characteristics of the Included Studies

Of the 29 studies included, 9 examined dogs with TLD surgically treated with concomitant PF, 14 studies were without PF, and 6 of them included both dogs with and without PF. Five out of those six studies were included in the meta-analysis (Table 2). One study was excluded from the meta-analysis as it could not provide sufficient data for the recurrent dogs. Most of them were retrospective studies (24), three were prospective studies/prospective case series study and two were clinical trials. Finally, data for 5457 dogs with thoracolumbar IVDH were collected and examined in terms of recurrence incidence.

### 3.3. Risk of Bias in the Included Studies

Twenty-seven cohort studies were evaluated in seven domains of bias due to confounding factors, selection of participants into the study, classification of intervention, deviations from intended interventions, missing data, measurement of outcome, and selection of the reported result as proposed by the guidelines. The majority of the included studies were evaluated as serious (16), while for 10 studies the evaluation grade was critical and for 1 was moderate (Figure 2 and Figure 3). The two clinical trials (one randomized, one blinded) were evaluated for risk of bias with the ROB2 tool in five domains: randomization process, deviation from intended intervention, missing outcome data, measurement of the outcome, and selection of the reported result. One was evaluated as low bias and the second as high bias, due to missing data (Figure 4 and Figure 5).

### 3.4. Characteristics of the Examined Population

Five thousand four hundred and fifty-seven canines with TL IVDH were examined and surgically decompressed in the 29 studies included in the SR. Dachshunds were the most affected breed with 2161 (39.60%) affected animals. In seven studies the population examined was: (a) only chondrodystrophic breeds [23,113,114]; (b) only Dachshunds [117,120]; (c) only large breed dogs [117]; and (d) only French bulldogs [123]. In one study the breeds were recorded only for the recurrent cases [107]. Twelve studies (*n* = 2593 dogs) [19,20,21,22,24,104,106,107,108,113,116,121] recorded the primary affected site, with the highest presentation of IVDH being at T12-13 (*n* = 484, 18.66%).

### 3.5. Characteristics of Prophylactic Fenestration

In total, 15 out of the 29 included studies have examined dogs with TL IVD surgically treated with concomitant PF (see in Table 2). Fourteen had full records for the number of dogs that underwent PF. In one study the number of dogs was not specified [122]. Overall, 1487 dogs out of the total 5457 dogs (27.25%) belonged to the intervention group. The types of prophylactic fenestration used were manual blade fenestration [19,20,21,23,24], power-assisted fenestration [21,24], PLDA [8,15,109,118], and chemonucleolysis [24].

### 3.6. Recurrence Incidence among the Surgically Treated Dogs

Out of 5457 dogs surgically treated for TL IVDH, available follow-up existed only for 5235 of them. The two author groups assessed the recurrence incidence in all the studies as established from our inclusion criteria. Some dogs experienced more than one incidence of recurrence. No recurrence incidence occurred in three of the included studies [20,110,120]. Another three did not provide full data records for the dogs that experienced a recurrent episode [19,122,123]. Five hundred and four dogs had at least one suspected or confirmed episode of recurrence of the total dogs examined (total: 5457 dogs). From the total recurrences, 105 were recorded in Dachshunds (20.83%). However, only in 11 studies out of 29 the breed of recurrent dogs was recorded [15,19,21,22,24,105,106,107,118,121,123]. No sufficient data were able to be collected to distinguish the recurrent Dachshunds that underwent PF from the included studies. Recurrent PF cases (total = 164, confirmed = 61, suspected = 103) represented 32.54% of the total 504 recorded recurrences (and 11.02% of the total PF cases (*n* = 1487). A total of 82 out of 164 incidences of recurrence in PF cases (50.0% of PF recurrences; 16.27% of total 504 recurrences) were found to occur at a site other than the PF treated area, and only in eight cases (4.88% of PF recurrences; 1.59% of total 504 recurrences) were located on the surgically treated area. In five studies [15,108,114,118,122] the exact site of the recurrence was not investigated and in one study [119] the group in which the relapsed cases occurred was not recorded. Table 3 illustrates a summary of the recurrence cases.

#### 3.6.1. Recurrence Incidence Cranially to Primary Surgically Treated Site 

The recurrence incidence cranially to the primary surgically treated site retrieved from seven studies is demonstrated in Table 4. Out of 504 dogs that experienced recurrence, 17 (3.37%) involved a disc more cranially than those surgically treated with decompression. Six of such recurrences occurred in the PF group of dogs and represented 3.66% of PF recurrent dogs (*n* = 164).

#### 3.6.2. Recurrence Incidences on the Caudal Lumbar Area (L3-L4-L5-L6)

Recurrence incidences on caudal lumbar area were recorded on six studies [19,21,24,107,116,121]. Of the 25 incidences, 10 were on L3-4, 12 on L4-5, and 3 on L5-6. In one study [107], there was one more caudal recurrence on L3-4. This was due to a previously operated disc, and it was excluded. These recurrences represent 4.96% of total recurrences (*n* = 504). Seventeen of this type of incidence occurred in the intervention group representing 68.0% of recurrences on the caudal lumbar area (*n* = 25) and 10.36% of the total incidences in PF group (*n* = 164). Results are shown in Table 5.

### 3.7. Secondary Outcome: Complications Associated with PF

Complications associated with PF were reported in 10 studies. The most common complications seen among the examined population were: discospondylitis (3 dogs); mild pneumothorax (4 dogs); hemorrhage or hematoma at the needle site in the dogs that underwent PLDA (10 dogs); exacerbation of the neurological deficits (14 dogs); abdominal wall weakness (4 dogs); and ataxia (7 dogs). In Table 6, complications are presented in detail.

### 3.8. Meta-Analysis for Recurrence Ratio between Dogs with and without PF

A meta-analysis was conducted to determine the exact ratio of recurrence between dogs with or without prophylactic fenestration. Data from 1406 dogs were collected out of five studies that met the inclusion criteria [19,21,24,104,105]. Overall, PF was performed on 1134 dogs and 272 dogs treated without PF. In total, 59 dogs experienced recurrence (PF *n* = 43, without PF *n* = 16). The heterogeneity within the five included studies was substantial and statistically significant (Tau^2^ = 1.26; Chi^2^ = 10.47; df = 4 (*p* = 0.03); I^2^ = 62%). The overall effect of the meta-analysis was statistically non-significant (*p* = 0.97). The results are demonstrated in Figure 6.

### 3.9. Meta-Analysis for Recurrence Ratio on Disc(s) Adjacent to Surgical Treated Area between Dogs with and without PF

A meta-analysis was conducted to demonstrate the ratio of recurrence on the adjacent disc to previously surgically treated dogs with or without PF. Data from three studies that met the inclusion criteria were collected [19,24,104]. Recurrence incidents on adjacent discs were recorded, all of them on PF group of dogs. Overall, 16 dogs experienced this type of recurrence. The results of this meta-analysis are presented in Figure 7. The heterogeneity among the three studies was substantial and statistically non-significant (Tau^2^ = 3.05; Chi^2^ = 4.53; df = 2 (*p* = 0.10); I^2^ = 56%). The overall effect on the difference in ratio between the two groups was statistically non-significant (*p* = 0.85).

## 4. Discussion

To our knowledge, this is the first systematic review and meta-analysis to robustly estimate the incidence of recurrence in dogs that underwent PF for TLD herniation. From the included studies, data about the initial treatment, the performance of PF, the incidence of recurrence, and the site of recurrence were extracted. Most of the studies were considered to be running a serious to critical risk of bias, with missing data being the key factor impacting the overall bias grade.

This SR can make a range of important assumptions. From the pool of the 5457 surgically treated dogs, the total number of recurrences was low at 9.23% (number of dogs (*n* = 504)). From the total recurrences (*n* = 504, suspected and confirmed) the percentage of canines previously fenestrated was relatively high (32.54%, *n* = 164). Relapses on previously prophylactically fenestrated discs were recorded only in eight cases in the SR. They represent only 4.88% of the total incidences in dogs that underwent PF (*n* = 164, suspected and confirmed). Eighty-two (50.0%) of PF recurrent cases had a confirmed recurrence in other than the fenestrated discs. In a published review about the prophylactic role of the fenestration in TL IVDH, a recurrence rate is referred to from 0–24.4% [1]. This finding is in line with the results of the present SR. The same question appears once again about the role of fenestration as a prophylaxis for future disc extrusions on adjacent and further discs. These recurrences may be correlated with vertebral instability as AF has a stabilizing effect on spine. A cadaveric study [43] supports that the combination of hemilaminectomy and fenestration produced the greatest degree of instability in the vertebral column, with fenestration being the most significant, single destabilizing factor. In an effort to quantify the incidence in the previously fenestrated dogs, a meta-analysis was conducted. The measured heterogeneity was substantial and significant, but the overall effect of PF on recurrence ratio was non-significant. At this point it is important to mention that the surgical technique used in the majority of these studies (with only one exception) was manual blade or power-assisted fenestration. Records for the type of the chosen fenestration technique performed in the recurrent cases were missing. Hence this review cannot draw safe conclusions about the incidence of recurrence according to the type of fenestration.

As noted above, 82 confirmed cases of recurrences were found in areas other than the primary PF treated area. Seventeen of those incidences (10.36% of recurrent PF cases) occurred in the caudal lumbar area (L3-4-5-6). Interestingly, the caudal lumbar area was the site of spontaneous incidence of IVDH in 3.7% and 7% of reported cases [1,2,24], which is less than the rate in this SR. It seems that PF may result in non-fenestrated adjacent disc herniation. Moreover, 6 out of 164 recurrences (3.66%) on dogs that underwent PF have occurred in a disc more cranial than the primarily treated discs. These results are in agreement with the widely held belief that prophylactic fenestration may decrease the recurrence in the disc that underwent treatment, but on the other hand, may result in other IVDH in discs located adjacent or further away from the treated site. Aikawa et al., 2012 [19] recommended that if there is a concern of possible vertebral instability, the vertebral segment should be stabilized, to minimize the possibility of recurrence in further discs. Considerations also exist for the iatrogenic nerve roots damage of the lumbar intumescence, during fenestration on the caudal area of L4-L5-L6 [26,107]. Only three studies met the criteria for a meta-analysis to correlate the incidence of recurrence in the adjacent discs between the surgically treated canines with or without concomitant PF. Once again, the heterogeneity was substantial but statistically non-significant and so was its overall effect.

In this systematic review, data were also collected about the breed and the site of presentation of IVDH. Dachshunds represented in the majority of the cases with IVDH. In the relevant literature, chondrodystrophic breeds and, more specifically, Dachshunds are at higher risk to experience at least one episode of IVDH during their life and also at higher risk of recurrence [1,13,26]. In this SR, the recurrent episodes of Dachshunds reflect 20.83% (*n* = 105) of the total recurrences. Although Dachshunds seem to be at a higher risk for recurrence than other breeds in this SR, it will be unreasonable to make this assumption as the breed of the recurrent dogs was reported in only 8 out of 27 studies (in three studies, there was no recurrence).

As for the complications related to the prophylactic fenestration, only 10 out of the 17 included studies where PF was performed recorded the incidence of complication postoperatively. Complications associated with the surgeon’s skills or with the chosen technique of fenestration (spinal cord damage, hematomas at the site of insertion, hemorrhages, pneumothorax, hemothorax, nerve root trauma) could be limited [19]. For other complications, such as abdominal wall weakness, discospondylitis, vertebral instability pneumothorax, and neurological deterioration postoperatively, further research is needed, apart from cadaveric studies, to show the actual clinical impact on dogs that underwent PF.

A significant limitation to this SR is that available studies that detect the recurrence incidences post prophylaxis of intervertebral discs by fenestration, were qualitative assessed to have serious to critical risk of bias. Authors also collected data from studies whose aim was not always to screen the incidence of the recurrence, but enough information was presented. The decision to include these studies was based on the relatively small pool of available studies. Relevant studies that were not written in English, or studies that were not available on electronic databases were excluded, which may have affected our results. The available literature includes retrospective studies, among which there was a high variability of: (1) the selected dogs (chondrodystrophic or not, small or large breeds, dogs with or without previous IVDH episodes, presence or not of deep pain perception on the examined canines), (2) intended intervention techniques of fenestration, and (3) follow-up periods. The main limitation among the included studies, which affected the bias risk of this SR, was that the postoperative follow-up was based on telephone contact with the owners and many of the relapsed incidences recorded were suspected but not confirmed. Authors believe that the low evidence bias was expected as 24 of the 29 studies were retrospective and the inclusion criteria and the outcome of each study differed. This was a serious limitation to the attempt to provide a definitive answer to the research question of this study.

Prophylactic fenestration is not consistently performed with individual variability as its effectiveness is still questioned. The complications associated with the procedure, the increase surgical and anesthesia time, postoperative morbidity, and the relative cost should be taken under consideration. This systematic review could not reach a safe conclusion about its contribution towards prophylaxis against recurrence in dogs suffering TLD herniation.

## 5. Conclusions

In conclusion, the question of whether PF decreases the rate of recurrence in 5457 cases remains unanswered. PF protects the treated discs from further herniation. However, the risk to the adjacent and further discs has not been determined. The role of the AF for the vertebral stability of the spine is indisputable and it is sacrificed during this procedure. It is clear to the authors that the risk of recurrence in discs not fenestrated prophylactically remains high. Unfortunately, the quality of evidence provided in this SR is low and prevents the authors from reaching valuable conclusions on the recurrence incidence in dogs surgically treated with concomitant PF for thoracolumbar disc herniation. The authors highlight the need for further studies, preferably clinical trials, to offer objective insights into the recurrence risk in adjacent discs following surgical treatment with PF.

## Figures and Tables

**Figure 1 animals-12-02601-f001:**
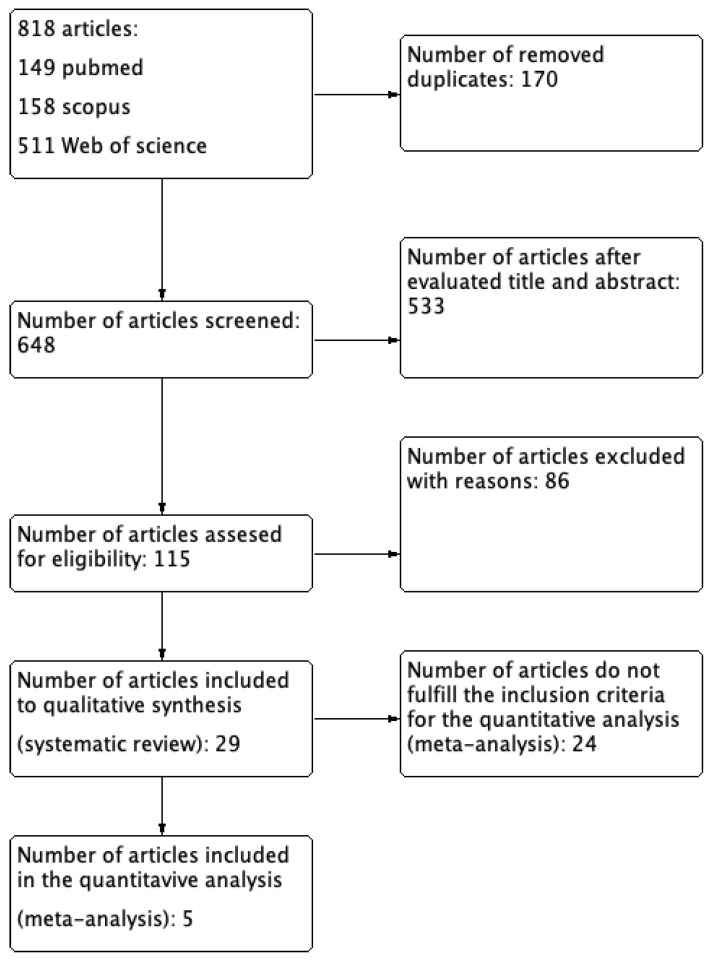
Flow chart of this systematic review.

**Figure 2 animals-12-02601-f002:**
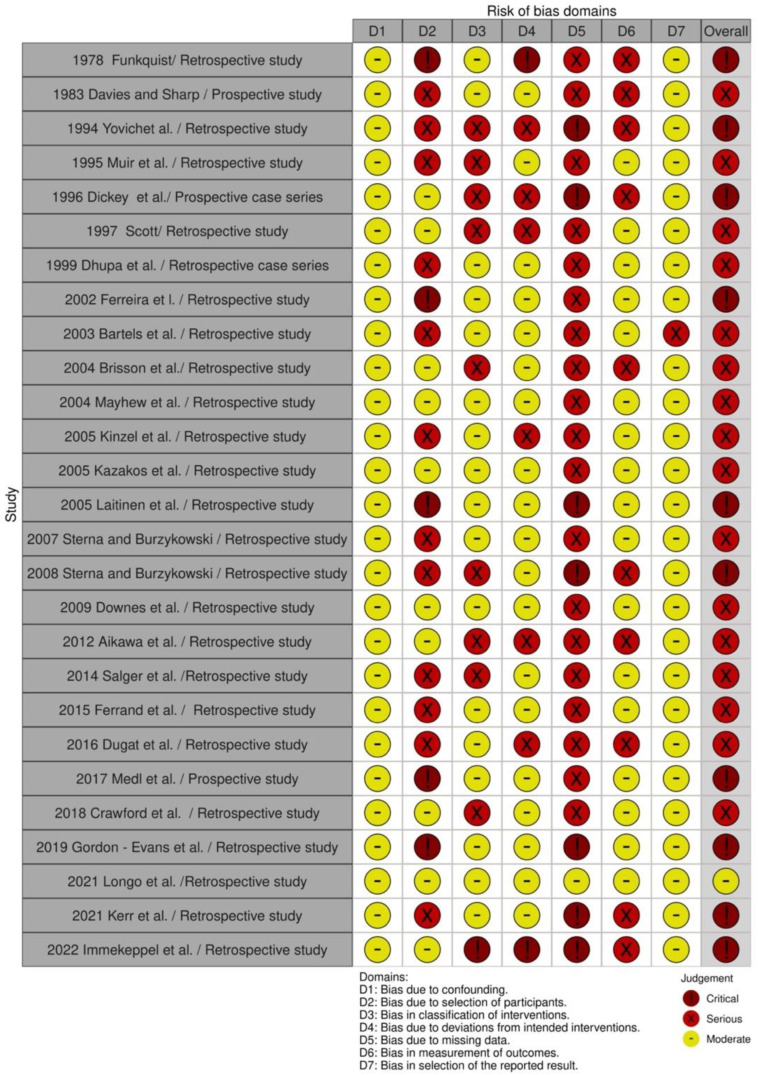
Traffic-light plot of cohort studies [8,15,19,20,21,22,23,24,85,104,105,106,107,108,109,110,111,112,113,114,115,116,117,118,119,120,121,122,123].

**Figure 3 animals-12-02601-f003:**
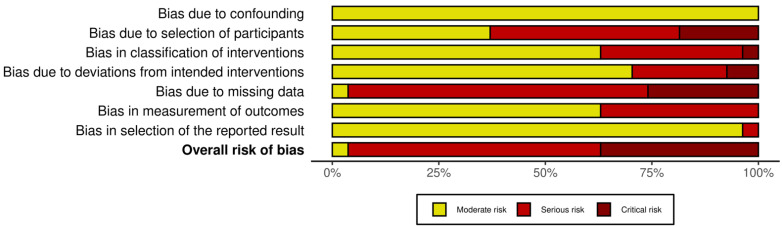
Summary plot of cohort studies.

**Figure 4 animals-12-02601-f004:**
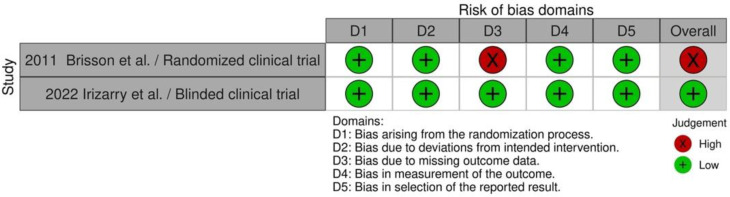
Traffic-light plot of clinical trials [8,21].

**Figure 5 animals-12-02601-f005:**
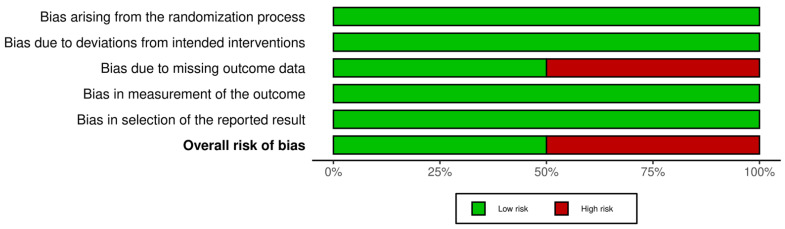
Summary plot of clinical trials.

**Figure 6 animals-12-02601-f006:**
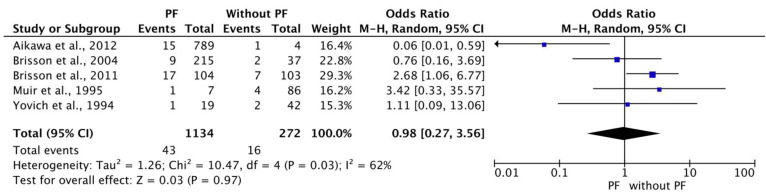
Meta-analysis for recurrence ratio (with prophylactic fenestrations; without prophylactic fenestration) [19,21,24,104,105].

**Figure 7 animals-12-02601-f007:**
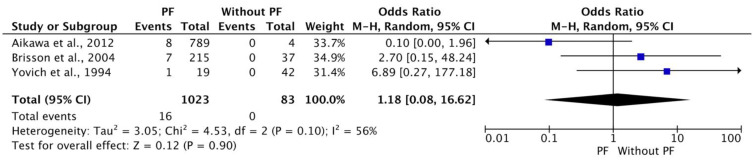
Meta-analysis for recurrence ratio on disc(s) adjacent to surgically treated area (with prophylactic fenestration; without prophylactic fenestration) [19,24,104].

**Table 1 animals-12-02601-t001:** PICO.

PICO	Inclusion Criteria
Population	Dogs with TL IVDH of all breeds and ages
Intervention/control	Surgically treated with and/or without PF
Primary outcome	Follow-up of the canines Recurrence of clinical signs at TL area (confirmed or suspected)
Secondary outcome	Complications related with PF
Others	Retrospective, prospective, case series, clinical trials studies Studies in English

P: population, I: intervention, C: control, O: outcome; PF: prophylactic fenestration; TL: thoracolumbar; IVDH: intervertebral disc herniation.

**Table 2 animals-12-02601-t002:** Studies included in the systematic review.

Author, Year	Type of Study	Number of Dogs	Number of Dogs without/with PF
Funkquist, 1978 [23]	Retrospective study	88	0/88
Davies and Sharp, 1983 [20]	Prospective study	34	0/34
Yovich et al., 1994 ^M^ [104]	Retrospective study	61	42/19
Muir et al., 1995 ^M^ [105]	Retrospective study	93	86/7
Dickey et al., 1996 [15]	Prospective case series	33	0/33
Scott, 1997 [106]	Retrospective study	40	40/0
Dhupa et al., 1999 [107]	Retrospective study	467	467/0
Ferreira, 2002 [108]	Retrospective study	71	0/71
Bartels et al., 2003 [109]	Retrospective study	277	0/277
Brisson et al., 2004 ^M^ [24]	Retrospective study	265	42/215
Mayhew et al., 2004 [22]	Retrospective study	229	229/0
Kazakos et al., 2005 [110]	Retrospective study	30	30/0
Kinzel et al., 2005 [111]	Retrospective study	331	162/169
Laitinen et al., 2005 [112]	Retrospective study	46	46/0
Sterna et al., 2007 [113]	Retrospective study	36	36/0
Sterna et al., 2008 [114]	Retrospective study	26	0/26
Downes et al., 2009 [115]	Retrospective study	28	28/0
Brisson et al., 2011 ^M^ [21]	Randomized clinical trial	207	103/104
Aikawa et al., 2012 ^M^ [19]	Retrospective study	793	34/759
Salger et al., 2014 [85]	Retrospective study	72	72/0
Ferrand et al., 2005 [116]	Retrospective study	107	107/0
Medl et al., 2017 [117]	Prospective study	57	57/0
Dugat et al., 2016 [118]	Retrospective study	303	0/303
Crawford et al., 2018 [119]	Retrospective study	53	53/0
Gordon-Evans et al., 2018 [120]	Retrospective study	32	32/0
Longo et al., 2020 [121]	Retrospective study	92	92/0
Immekeppel et al., 2021 [122]	Retrospective study	1501	1422/23 confirmed (56 of unspecified medical history)
S. Kerr et al., 2021 [123]	Retrospective study	55	33/22
Irizarry et al., 2022 [8]	Blinded control study	30	0/30

**^M^** Studies included in the meta-analysis; PF: prophylactic fenestration.

**Table 3 animals-12-02601-t003:** Summary of the recurrent cases.

Author/Year	Recurrent/Surgically Treated Dogs	Confirmed (without/with PF)	Suspected (without/with PF)
Funkquist et al., 1978 [23]	14/88	7 (0/7)	7 (0/7)
Davies et al., 1983 [20]	0/34	-	-
Yovich et al., 1994 [104]	8/61	3(2/1)	5 (3/2)
Muir et al., 1995 [105]	5/93	5 (4/1)	-
Dickey et al., 1996 [15]	5/33	2 (0/2)	3 (0/3)
Scott, 1997 [106]	5/40	1(1/0)	4 (4/0)
Dhupa et al., 1999 [107]	30/467	30 (30/0)	-
Ferreira et al., 2002 [108]	9/71	-	9 (0/9)
Bartels et al., 2003 [109]	9/277	8 (0/8)	1 (0/1)
Brisson et al., 2004 [24]	29/265	11 (2/9)	18 (5/13)
Mayhew et al., 2004 [22]	44/229	20 (20/0)	24 (24/0)
Kazakos et al., 2005 [110]	0/30	-	-
Kinzel et al., 2005 [111]	2/331	2 (2/0)	-
Laitinen et al., 2005 [112]	2/46	1 (1/0)	1 (1/0)
Sterna et al., 2007 [113]	8/36	3 (3/0)	5 (5/0)
Sterna et al., 2008 [114]	5/26	0	5 (0/5)
Downes et al., 2009 [115]	2/28	2 (2/0)	-
Brisson et al., 2011 [21]	55/207	24 (17/7)	31 (18/13)
Aikawa et al., 2012 [19]	81/793	15 (1/14)	66 (not recorded)
Salger et al., 2014 [54]	11/72	3 (3/0)	8 (8/0)
Ferrand et al., 2015 [116]	25/107	10 (10/0)	15 (15/0)
Dugat et al., 2016 [118]	60/303	11 (0/11)	49 (0/49)
Medl et al., 2017 [117]	7/57	5 (5/0)	2 (2/0)
Crawford et al., 2018 [119]	19/53	-	19 (19/0)
Gordon-Evans et al., 2019 [120]	0/32	-	-
Longo et al., 2021 [121]	33/92	8 (8/0)	25 (25/0)
Immekeppel et al., 2021 [122]	10/1501	-	10 (not recorded)
Kerr et al., 2021 [123]	24/55	-	24 (not recorded)
Irizarry et al., 2022 [8]	2/30	1 (0/1)	1 (0/1)

PF: prophylactic fenestration.

**Table 4 animals-12-02601-t004:** Recurrence of intervertebral disc herniation cranially to the primary surgically treated site.

Author/Year	Decompressive Surgical Site(s)	IVD Space of Recurrence
Yovich et al., 1994 [104]	L1-5	T13-L1 ^+^
Muir et al., 1995 [105]	T13-L4	T11-12 ^+^
Scott, 1997 [106]	T12-13	T11-12
Dhupa et al., 1999 [107]	T13-L1 T12-13 * T13-L1 * L3-4 L1-2 L2-3	T11-12 T11-12 * T11-12 * T11-12 T13-L1 T13-L1
Sterna et al., 2007 [113]	T12-13	L1-2
Brisson et al., 2004 [24]	L1-2	T11-12 ^+^
Aikawa et al., 2012 [19]	T11-L2	T10-11^+^

* Two recurrence episodes recorded, ^+^ dogs in group with prophylactic fenestration.

**Table 5 animals-12-02601-t005:** Number and sites of recurrences on IVD spaces on caudal lumbar area.

Author/Year	L3-4	L4-5	L5-6
Dhupa et al., 1999 [107]	3	2	
Brisson et al., 2004 * [24]	1	5	
Downes et al., 2009 [115]	1		
Brisson et al., 2011 * [21]		3	1
Aikawa et al., 2012 * [19]	5	2	
Longo et al., 2021 [121]			2

* Recurrences in prophylactically fenestrated dogs.

**Table 6 animals-12-02601-t006:** Complications associated with PF (number of dogs).

Author/Year	Complications
Funkquist, 1978 [23]	Spinal cord damage (3); discospondylitis (1)
Bartels et al., 2003 [109]	Mild pneumothorax (1); abscess at needle insertion site (1); proprioceptive deficits (3); discospondylitis (1)
Dickey et al., 1996 [15]	Minimal hemorrhages from needle insertion
Brisson et al., 2004 [24]	Mild pneumothorax (1); fatal hemothorax (1)
Kinzel et al., 2005 [111]	Pneumothorax (1); Discospondylitis (1)
Brisson et al., 2011 [21]	Remaining disc material (1); vertebral sinus hemorrhage (3); vertebral artery hemorrhage (7); nerve root trauma causing abdominal weakness (4); curette tip breakage within disc space (1)
Aikawa et al., 2012 [19]	Vertebral instability or subluxation at the site of hemilaminectomy with PF (3)
Dugat et al., 2016 [118]	Hematoma formation at needle insertion site (1); post-surgical ataxia (6); soreness along the back (4); greater pain than expected (2); reluctance to move (2); pain on palpation in the spinal region (1); licking and chewing at the left hind paw (1)
Sterna et al., 2008 [114]	Exacerbated neurological deficits post-surgery (1); wound infection (1); death due to uraemia (1)
Immekeppel et al., 2022 [122]	Exacerbated neurological deficits (10) (intervention group not specified)

## Data Availability

Data are contained within the article.

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
