# Peer review of "Prophylactic Effect of Fenestration on the Recurrence of Thoracolumbar Intervertebral Disc Disease in Dogs"

_animals, 2022, doi:10.3390/ani12192601_

Round 1

Reviewer 1 Report

Thank you for submitting this systematic review, your manuscript revolves around a very important topic, the usefulness of prophylactic fenestration in the prevention of reoccurrence of thoracolumbar IVDH in dogs. Despite the fact this procedure is commonly performed, the exact benefit of it is still debatable. The aim of your review is to quantitatively and qualitatively assess the published literature to see if evidence can be found regarding the benefit of prophylactic fenestration in TL IVDD.

Overall the manuscript offers a good summary of what has been publish on this topic so far, however it will be critical to improve some aspects, such as the material and method section, and organise the results so that they are easier to follow. I appreciate the challenges of performing this SR and the manipulation of a vast amount of data but this makes it even more important to be very clear and consistent in the presentation of the results. The discussion contains some methods that appear here for the first time (279-281), inaccuracies (295 …”AF has stabilising effects on the spinal cord” and unsupported statements (309-312 or 335-336) and needs overall some work so that it is focused on the results.

Specific comments

Title:

The title is clear and reflect the topic under investigation.

Abstract:

The abstract is concise and clear

Introduction:

Line 43, ref 8 is not appropriate, please check.

Line 59-60 needs a reference

Line 63-65; needs re-phrasing

Line 73 pelvic limbs preferable over hind limbs; the sentence in the parentheses needs reviewing root, should read nerve root, also check spellings)

Methods:

As stated above the methods need to be more specific and clearly state the reasons for including and excluding studies not just at the initial stage but mostly at the refining stage when from 115 articles you reduced to 29. I would also suggest clarifying in the methods what elements you are assessing/comparing, this only becomes apparent in the results. It would not be possible to re-produce your study based on your current methods.

Results:

I found this section very, very difficult to follow and I would advise restructuring to clearly deliver the information you have obtained in your review.

Line 149 this paragraph belongs to the methods

Line 199 where does the n 15 comes from? Is this 9+6…I had to look for it and not sure this is the correct conclusion. It is very confusing to keep skipping form number of studies to number of dogs, this need to be clearly highlighted some way or other.

Also numbers need to be expressed as cyphers unless they are at the start of a sentence

Line 203 PDLA this acronymous is not explained anywhere that I can find. If the journal allows a table with abbreviations would be useful.

Line 205 numbers need to be expressed as cyphers unless they are at the start of a sentence

Line 210, 504 does this include the control group? It is neve completely clear when you refer to the whole population and when to the control/intervention group.

Line 220 “on it” does this refer to the fenestrated disc?

Line 225 please clarify why you investigated this aspect separately

Out of the 504 dogs with recurrences how many had repeated images to confirm the site?

Line 232 clarify why you investigated this region separately

Line 255 “outcome” does this refer to the heterogenicity assessment?

Line 256 “the overall effect” does this now refer to the PF?

Line 261 “adjacent” to the IVDH or PF?

Line 263 why only 3 studies?

Discussion

Line 279-281 this part belongs to the methods

Line 280 “operated”, should this be fenestrated?

Line 291 “contributes” not sure what you mean here

Line 295 “spinal cord” is wrong

Line 309-312 as stated above this is not supported by your findings

Line 315-318 not sure about the relevance of this sentence

Line 335-336 also not sure about this sentence and its relevance

Table 4, 5 are not very useful in the current format and do not add to the text

Figure 2, 3, 4 and 5 also do not add to the text

Reviewer 2 Report

Dear authors,

The review covers a large number of publications, clinical cases, approaches, through several years. It is well written although some extra English editing is needed. As for example on line 46 the word "disc" is wrongly written.

There are two points where the review could be improved. First, the surgical approach used, briefly commented on point 3.5. However there is no analysis of recurrence vs success of outcome depending on type of surgical approach used. As overall, it cannot be concluded if PF is needed or not to avoid recurrence, but the way this is done could also play a role in the outcome of it, but there is no analysis on outcome according to surgical approach. And second, it is mention in the introduction the differences between surgeons and neurologist, on how uses more PF and who less. Lines 47 to 51. I assume of the many reviewed articles, some most come from a surgeons report and others from neurologists reports? It would also be worth to note if the outcome is worst or better depending on the specialty of the treating veterinarian. However, nothing is mentions about it.

Overall is a well presented review, that could benefit from some extra results and analysis.

Round 2

Reviewer 1 Report

Dear Authors,

thank you for taking the time to respond to the previous comments and for the hard work to improve the manuscript. I feel the the manuscript is much easier to read now and the layout of the results is much easier to follow. 

There are still several typos and incomplete sentences, some I have highlighted in the text.  I have also embedded small comments in the text (see attached file)

Regarding figures 2 to 5, I still don't think they add to the manuscript but will leave the final decision regarding keeping/removing up to the editor.

Author Response

Dear Reviewer,

First of all, we would like to thank you for editing the text and the time you spent on it. Your help was instrumental in optimizing this manuscript.

1. We would like to mention that many times during the transfer of the document via email, many phrases loose the space between them. We have noticed this problem already from the previous time and we have informed the editor. That was why we uploaded the paper both in .pdf and a .doc format. In case that you will find again few typos, probably it will be the same problem.

2. Please check all the text changes / rephrases following your recommendations.

3. As for the paragraph “3.1 Characteristics of the excluded studies”, as it is part of the guidelines of the Systematic Review Protocol forAnimal Intervention Studies (SYRCLE) we believe that it should retained in the text. Although, if you agree we could leave this decision to the editor.

4. Please check in paragraph “3.7. Secondary outcome: Complications associated with PF” the number addition as you have recommended.

5. Please check the highlights (yellow), we have changed the acronyms from IVDD to IVDH to be more clear and less frustrated to the readers.

6. Finally, please check the rephrase on line 56.
